TROPICAL DISEASES

# Hepatitis E as a cause of adult hospitalization in Bangladesh: Results from an acute jaundice surveillance study in six tertiary hospitals, 2014-2017

**Repon C. Paul** [1,2]*, **Arifa Nazneen**[1], **Kajal C. Banik**[1], **Shariful Amin Sumon**[1], **Kishor K. Paul** [1], **Arifa Akram** [3], **M. Salim Uzzaman**[3], **Tahir Iqbal**[4], **Alexandra Tejada-Strop**[4], **Saleem Kamili**[4], **Stephen P. Luby**[5], **Heather F. Gidding**[6], **Andrew Hayen**[7], **Emily S. Gurley**[1,8]

1 icddr,b, Dhaka, Bangladesh, 2 School of Public Health and Community Medicine, UNSW Medicine, Sydney, Australia, 3 Institute of Epidemiology, Disease Control and Research, Government of the People's Republic of Bangladesh, 4 Division of Viral Hepatitis, Centers for Disease Control and Prevention, Atlanta, Georgia, United States of America, 5 Infectious Diseases and Geographic Medicine, Stanford University, Stanford, California, United States of America, 6 Faculty of Medicine and Health, University of Sydney, Sydney, Australia, 7 Australian Centre for Public and Population Health Research, University of Technology Sydney, Sydney, Australia, 8 Johns Hopkins Bloomberg School of Public Health, Baltimore, Maryland, United States of America

* reponpaul@yahoo.com

**Data Availability Statement:** All relevant data are within the manuscript and its Supporting Information files.

## Abstract

In the absence of reliable data on the burden of hepatitis E virus (HEV) in high endemic countries, we established a hospital-based acute jaundice surveillance program in six tertiary hospitals in Bangladesh to estimate the burden of HEV infection among hospitalized acute jaundice patients aged ≥14 years, identify seasonal and geographic patterns in the prevalence of hepatitis E, and examine factors associated with death.

We collected blood specimens from enrolled acute jaundice patients, defined as new onset of either yellow eyes or skin during the past three months of hospital admission, and tested for immunoglobulin M (IgM) antibodies against HEV, HBV and HAV. The enrolled patients were followed up three months after hospital discharge to assess their survival status; pregnant women were followed up three months after their delivery to assess pregnancy outcomes. From December'2014 to September'2017, 1925 patients with acute jaundice were enrolled; 661 (34%) had acute hepatitis E, 48 (8%) had hepatitis A, and 293 (15%) had acute hepatitis B infection. Case fatality among hepatitis E patients was 5% (28/ 589). Most of the hepatitis E cases were males (74%; 486/661), but case fatality was higher among females—12% (8/68) among pregnant and 8% (7/91) among non-pregnant women. Half of the patients who died with acute hepatitis E had co-infection with HAV or HBV. Of the 62 HEV infected mothers who were alive until the delivery, 9 (15%) had miscarriage/stillbirth, and of those children who were born alive, 19% (10/53) died, all within one week of birth. This study confirms that hepatitis E is the leading cause of acute jaundice, leads to hospitalizations in all regions in Bangladesh, occurs throughout the year, and is associated with considerable morbidity and mortality. Effective control measures should be taken to

**Funding:** This research was funded by Centers for Disease Control and Prevention (CDC), USA. icddr, b is also grateful to the Government of Bangladesh, Canada, Sweden and the UK for providing core/ unrestricted support. Support to Paul RC was given by the UIPA (University International Post graduate award) scholarship from UNSW. The funders had no role in study design, data collection and analysis, decision to publish, or preparation of the manuscript.

**Competing interests:** The authors have declared that no competing interests exist.

reduce the risk of HEV infections including improvements in water quality, sanitation and hygiene practices and the introduction of HEV vaccine to high-risk groups.

## Author summary

In the absence of reliable surveillance data on the burden of hepatitis E in endemic countries, we conducted a hospital-based acute jaundice surveillance study over a two and a half year period in six tertiary hospitals in Bangladesh. The study confirms that HEV infections occur throughout the year, and is a major (34%) cause of acute jaundice in tertiary hospitals in Bangladesh. Three-quarters of the acute hepatitis E cases were male, and HEV infection was higher among patients residing in urban areas than patients in rural areas (41% vs 32%). The overall case fatality rate of acute HEV infections in hospitals was 5%, but was higher among pregnant women (12%). Hepatitis E patients who died were more likely to have co-infection with HAV or HBV than the HEV infected patients who did not die. Fifteen percent of HEV infected mothers had miscarriage/stillbirth. Of the children who were born alive, 19% died, all within one week of birth. Considering the high burden of hepatitis E among hospitalized acute jaundice patients, Bangladesh could take control measures to reduce this risk including improvements in water quality, sanitation and hygiene practices and the introduction of hepatitis E vaccine in high-risk areas.

## Introduction

Hepatitis E virus (HEV) infection causes inflammation of the liver and is an important cause of acute jaundice, especially in resource-poor countries, where fecal contamination of drinking water is common [1–3]. HEV genotypes 1 and 2 are predominantly spread by the fecal-oral route [4]. By contrast, HEV infection in high-income countries occasionally occurs as a result of zoonosis of genotypes 3 and 4, being transmitted via exposure to wild animals, and consumption of undercooked pork or game meat [5,6]. HEV genotypes 1 and 2 in high-income countries are generally limited to travellers to hepatitis E endemic countries [7,8].

HEV can cause sporadic cases and small outbreaks [9], but is often responsible for epidemics in Asia and many parts of Africa [1,10–12]. The most common clinical feature of symptomatic HEV infection is jaundice; it is symptomatically indistinguishable from other causes of acute viral hepatitis [13]. Symptomatic illness due to HEV infection is infrequent in children [14,15]; the infection primarily affects young adults, and is generally mild and self-limiting [2,16,17]. However, the case fatality among HEV-infected pregnant women has been reported to be as high as 6–30% [2,18–22]. Miscarriages, stillbirths and neonatal deaths are frequently observed among HEV-infected pregnant women [20,23–25].

The World Health Organization (WHO) identified viral hepatitis as a global health problem and set a goal to eliminate viral hepatitis by 2030 [26]. Even though an effective low-cost hepatitis E vaccine is available [27,28], the vaccine is not recommended for general use in endemic countries, largely because the burden of disease is not quantified and so it is unclear if use of the vaccine is appropriate and cost effective [29]. Hepatitis E disease surveillance data are limited to a few high-income countries [30,31]; most of the data from endemic countries are limited to disease outbreak and case series reports.

While hospital-based surveillance represents only the burden of severe HEV disease, it can provide valuable insights, especially regarding the contribution to mortality among severe

acute jaundice patients. Patients with acute jaundice commonly present to hospitals in many resource-poor hepatitis E endemic countries, including Bangladesh [32–34]. However, there are few studies in hepatitis E endemic countries that systematically enrolled patients with acute jaundice to estimate the burden of acute HEV infection in hospital settings, especially the case fatality rate and delivery outcomes of pregnant women [35,36]. Previous hospital-based studies in Bangladesh suggest that most of the cases of acute jaundice (22–64%) were due to HEV infection, however, those studies were limited to a single study site or a short study period which may not be a representative estimate of the burden of hepatitis E among all hospitalized acute jaundice patients in Bangladesh [34,37,38].

The International Centre for Diarrhoeal Disease Research, Bangladesh (icddr,b) in collaboration with the Institute of Epidemiology, Disease Control and Research (IEDCR) of the Government of Bangladesh, and the United States Centers for Disease Control and Prevention (CDC), conducted a hospital-based acute jaundice surveillance study in six tertiary hospitals located in five regions of Bangladesh. The main objectives of this study were to estimate the prevalence of HEV infection among acute jaundice patients aged ≥14 years admitted to tertiary hospitals in Bangladesh and estimate the case-fatality among these patients. Secondary objectives included describing seasonal and geographic trends in the prevalence of HEV disease, identifying factors associated with mortality, measuring pregnancy outcomes and neonatal mortality of children born to mothers with acute HEV infection.

## Methods

### Surveillance sites and methods

Hospital-based acute jaundice surveillance was established in six tertiary hospitals, located in five of seven divisions (the second highest level of geographic and administrative areas) in Bangladesh (Fig 1). Five of these were government teaching hospitals and one was a private teaching hospital. Severely ill patients are generally referred to these tertiary hospitals from lower level hospitals located in the adjacent districts of the surveillance hospitals. The surveillance started between 6 December 2014 and 27 March 2015 (depending on the hospital) and continued until 30 September 2017 in all six sites.

We recruited a physician in each of the six hospitals from the existing staff to oversee the surveillance activities. A field assistant from icddr,b in each hospital assisted the surveillance physicians to collect patient information and blood specimens, and to store and transport samples to the laboratory in Dhaka. Every weekday morning, the surveillance physicians visited the obstetrics and gynaecology wards and the adult medicine wards of the hospitals and reviewed admission records from the previous day to determine if any patients aged ≥14 years met the case definition of acute jaundice. Due to resource constraints, we were unable to conduct surveillance in all wards in each hospital; therefore we restricted enrolment to adult medicine, obstetrics and gynaecology wards (where patients aged > 14 years are admitted) as children have been reported to have less severe presentations of HEV infection [14,15]. Patients with acute jaundice admitted on Friday were enrolled on the next working day, as study staff did not work on Fridays. Acute jaundice was defined as new onset of either yellow eyes or skin for less than 3 months, and continuing on the day of admission. The field assistants also visited the intensive care unit (ICU) every weekday to ensure that no acute jaundice cases were overlooked. If anyone met the case definition of acute jaundice, the surveillance physicians recorded enrolled patients' illness history and relevant clinical information (S1 Table) along with basic demographic information and contact information in a handheld computer using a structured case investigation form. The surveillance physicians also collected a 5 ml blood specimen for laboratory testing.

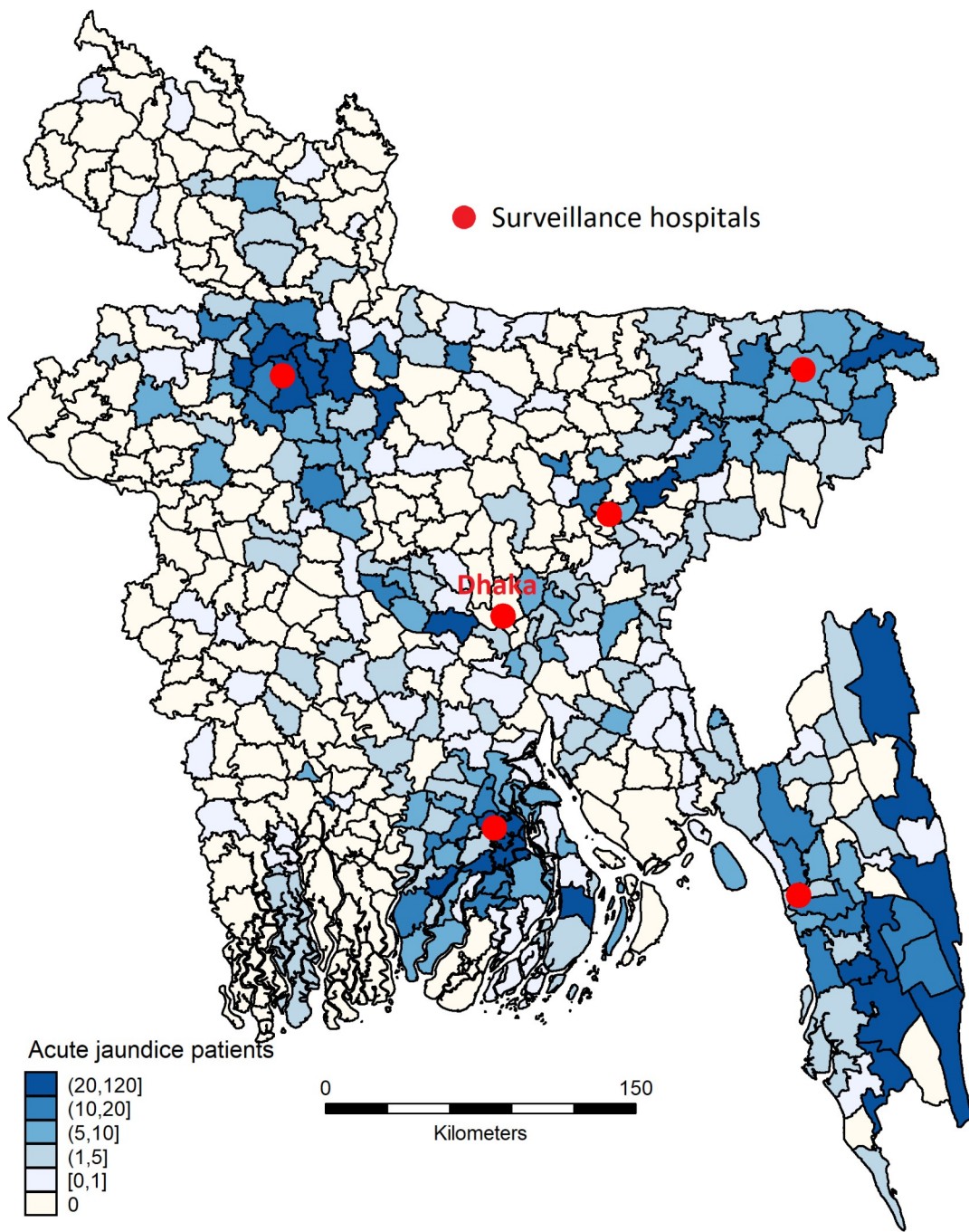

**Fig 1. Location of acute jaundice surveillance hospitals.** Map of Bangladesh showing the location of surveillance hospitals and number of enrolled patients with acute jaundice by sub-district during December 2014- September 2017. Map was created using the SPMAP module for Stata 14 (StataCorp) [39].

## Sample processing and laboratory testing

On the same day of sample collection, the field assistants centrifuged the blood specimen and prepared three aliquots of sera. One aliquot containing about 400μl of serum was sent to the hospital laboratory the same day of sample collection to test for bilirubin and glutamic-pyruvic

transaminase to assess liver function. Two aliquots of serum were stored at the surveillance hospitals in a liquid nitrogen dry shipper and transported to the laboratory of IEDCR in Dhaka every two weeks. The samples were kept in a freezer at IEDCR at -80˚C until they were tested. The samples were tested for IgM and IgG antibodies to HEV using enzyme-linked immunosorbent assay (ELISA) kits manufactured by Wantai, China (Beijing Wantai Biologic Pharmacy Enterprise Co., Ltd, Beijing, China) according to the manufacturer's instructions. Acute HEV infection was defined as a positive test result for anti-HEV IgM antibodies. We also tested all samples for hepatitis A virus IgM antibodies (anti-HAV IgM) (a marker of HAV infection) and hepatitis B virus surface antigen (HBsAg) (a marker of acute or chronic HBV infection). The samples that were positive for HBsAg were also tested for hepatitis B virus core IgM antibodies (anti-HBc IgM; a marker of acute HBV infection). Chronic HBV infection was defined as a positive test result for HBsAg but negative test result for anti-HBc IgM. HBsAg negative samples are generally negative for anti-HBc IgM [40], therefore, only the HBsAg positive samples were tested for anti-HBc IgM. ELISA test kits manufactured by DiaSorin, Italy were used for testing anti-HAV IgM, HBsAg and anti-HBc IgM.

We sent a subset of samples (collected between December 2014 and December 2016) to the Division of Viral Hepatitis Laboratory of the US Centers for Disease Control and Prevention (CDC), Atlanta, USA to test for HEV RNA. All samples from patients with onset within three weeks of the blood draw were selected for RNA testing. For other patients, a random sample of 20% of anti-HEV IgM positive samples, 20% of anti-HEV IgG positive samples, 20% of samples negative for both IgM and IgG were chosen to test for HEV RNA. Serum samples were tested for HEV RNA using a quantitative real-time reverse transcriptase polymerase chain reaction (PCR) assay, capable of detecting HEV genotypes 1–3 with a limit of detection of 25 IU/ml, targeting a 69-bp fragment of open reading frame (ORF) 3 of HEV genome [41]. For quality assurance, the samples that were sent to CDC were also retested for anti-HEV IgM using the same ELISA kit as was used at the laboratory in Bangladesh.

### Patient follow-up

All enrolled patients were followed up during their hospitalization by the surveillance physicians to monitor the outcomes of the illness episode and the pregnancy outcomes for pregnant patients. If the pregnancy ended during hospitalization and the outcome was a live birth, the surveillance physician examined the newborn to collect clinical information and note any signs or symptoms of jaundice. All enrolled patients and the newborns were followed up 3 months post hospital discharge to ascertain their vital status. If a pregnant woman with jaundice was released from the hospital before the end of her pregnancy, the field assistants followed up with her by phone one week after the expected date of delivery and three months after the date of delivery to check on the health of the mother and the newborn (S2 Table). For patients who did not have a phone, the field assistants visited their home to follow-up. Follow up three months post hospital discharge was not possible for the patients who were admitted to surveillance hospitals after June 30, 2017, as the surveillance ended on September 30, 2017.

### Data analysis

We calculated the prevalence of acute HEV infection among patients with acute jaundice in each surveillance hospital and overall. Seroprevalence of acute hepatitis E was also calculated by patients' demographic characteristics and rural-urban area of residence. We compared the signs and symptoms during illness, pregnancy complications, pregnancy outcome and case fatality between acute jaundice patients with and without evidence of acute HEV infection. We examined potential risk factors of death among patients with acute hepatitis E including co-

infection with other hepatitis viruses (HAV, HBV). Comparison between categorical variables was performed using the chi-squared test or Fisher exact test where appropriate and comparison between continuous variables (non-normal distribution) was performed using the Wilcoxon rank sum test. The 95% confidence interval for proportions was calculated by using the Wilson method for a binomial distribution [42]. We examined the monthly trends in the number of patients with anti-HEV IgM antibodies. We also calculated the proportion of patients positive for anti-HEV IgM and anti-HEV IgG by duration of illness at the time of admission to hospital. The onset of illness was defined as the date when any sign and symptom developed, including experiencing yellow eyes or skin or nausea/vomiting, or also fever, anorexia, abdominal pain, melaena, or unconsciousness. Analyses were conducted in Stata 14 (StataCorp).

## Human subjects

The surveillance physicians sought consent from the patients, or their guardians in the case of severely ill patients, to enrol them in the study. Written informed consent was obtained from patients aged over 17 years. For patients aged between 14 and 17 years, written assent was taken from the patients as well as obtaining written consent from their parents or guardians. The study protocol was reviewed and approved by the institutional review board of the icddr,b (Protocol # PR-14060). US CDC involvement in the study did not constitute engagement in human subjects research and therefore, CDC relied on appropriate IRB or ethics committee approval of engaged institutions.

## Results

### Patient characteristics

In the six surveillance hospitals, a total of 2,091 patients aged ≥14 years met the case definition of acute jaundice during December 2014 to September 2017. Of them, 1,925 (92%) agreed to be enrolled in the study and provided a blood specimen for laboratory testing (S1 Fig). Of the enrolled patients, 1,314 (68%) were male and 958 (50%) were aged between 14 and 29 years. Of the female patients, 192 (31%) were pregnant. Patients admitted to the surveillance hospitals resided in 56 out of 64 districts in Bangladesh; however, a higher number of patients lived near surveillance hospitals (Fig 1).

### Laboratory testing and HEV seroprevalence

Among the 1,925 enrolled patients, 661 (34%) had IgM antibodies detected against HEV (acute HEV infection) and 652 (99%) of these also had IgG antibodies against HEV. Overall, 1,036 (54%) patients had detectable anti-HEV IgG antibodies. There were 148 (8%) patients who were positive for anti-HAV IgM antibodies and 663 (34%) who were positive for HBsAg. Of the 663 HBsAg positive patients, 293 (15% of all patients) had acute HBV infection. Among the 661 acute HEV patients, 9 (1%) also had acute HAV infection, 15 (2%) had acute HBV infection and 132 (20%) had chronic HBV infection.

Of the 661 patients with acute HEV infection, 483 (73%) resided in rural areas, 486 (74%) were male; their median age was 25 years (IQR: 20–33). Acute hepatitis E prevalence varied across the hospital sites, ranging from 17% in Sylhet to 61% in Dhaka (Table 1). Among the patients with acute jaundice, those who were male, aged between 20 to 39 years, resided in urban areas, educated, or whose monthly family expenditure was >15,000 takas (US$188) were more likely to have acute HEV infection (p<0.001). Acute HEV infection was higher among the pregnant women with acute jaundice than the non-pregnant women with acute jaundice (39% vs. 24%, p<0.001). The higher rate of acute HEV infection among educated and

**Table 1. Hepatitis E serological test results by demographic characteristics of patients with acute jaundice in six tertiary hospitals in Bangladesh, December 2014–September 2017.**

| Characteristics | Number of patients | Anti-HEV IgM Positive | | P-value[a] |
|---|---|---|---|---|
| | | **n** | **Percent (95% CI)** | |
| *Total patients* | *1925* | *661* | *34 (32–37)* | |
| Hospital sites | | | | *<0.001* |
| Bogra | 405 | 79 | 20 (16–24) | |
| Barisal | 366 | 174 | 48 (42–53) | |
| Kishoregonj | 200 | 36 | 18 (13–24) | |
| Chittagong | 435 | 160 | 37 (32–42) | |
| Sylhet | 237 | 40 | 17 (12–22) | |
| Dhaka | 282 | 172 | 61 (55–67) | |
| Sex | | | | *<0.001* |
| Male | 1314 | 486 | 37 (34–40) | |
| Female | 611 | 175 | 29 (25–32) | |
| Pregnancy status of women[b] | | | | *<0.001* |
| Non-Pregnant | 418 | 101 | 24 (20–29) | |
| Pregnant | 192 | 74 | 39 (32–46) | |
| Age in years, Median (min, max) | 29 (14, 72) | | 25 (15, 60) | |
| 14–19 | 341 | 124 | 36 (31–42) | *<0.001* |
| 20–29 | 617 | 294 | 47 (44–52) | |
| 30–39 | 349 | 142 | 41 (35–46) | |
| 40–49 | 236 | 65 | 28 (22–34) | |
| 50–59 | 184 | 28 | 15 (10–21) | |
| 60+ | 188 | 8 | 4 (2–8) | |
| Residence | | | | *<0.001* |
| Rural | 1494 | 483 | 32 (30–35) | |
| Urban | 431 | 178 | 41 (37–46) | |
| Education | | | | *<0.001* |
| None | 383 | 63 | 16 (13–21) | |
| Class 1–5 | 574 | 175 | 31 (27–34) | |
| Class 6–11 | 644 | 259 | 40 (36–44) | |
| Class 12 or more | 324 | 164 | 51 (45–56) | |
| Monthly household expenditure in Bangladeshi taka[c] | | | | *<0.001* |
| < 5000 (US$ 62) | 270 | 48 | 18 (13–23) | |
| 5000–9,999 (US$ 63–125) | 717 | 214 | 30 (27–33) | |
| 10,000–14,999 (US$ 126–187) | 399 | 142 | 36 (31–41) | |
| ≥ 15,000 (US$188) | 328 | 138 | 42 (37–48) | |

[a] chi-squared test

[b] Percentage was calculated among female patients

[c] Monthly household expenditure was unknown for 211 patients

Anti-HEV IgM = Immunoglobulin M antibodies against HEV

higher income group patients with acute jaundice in the study hospitals was consistent in the analysis by rural-urban area of residence (S3 Table).

Patients who were admitted to hospital between three and four weeks of onset of illness had the highest anti-HEV IgM positivity (46%; Fig 2). The proportion who were anti-HEV IgM positive declined with time after four weeks of onset of illness ($\chi 2$ for trend: 8.1; $p < 0.01$).

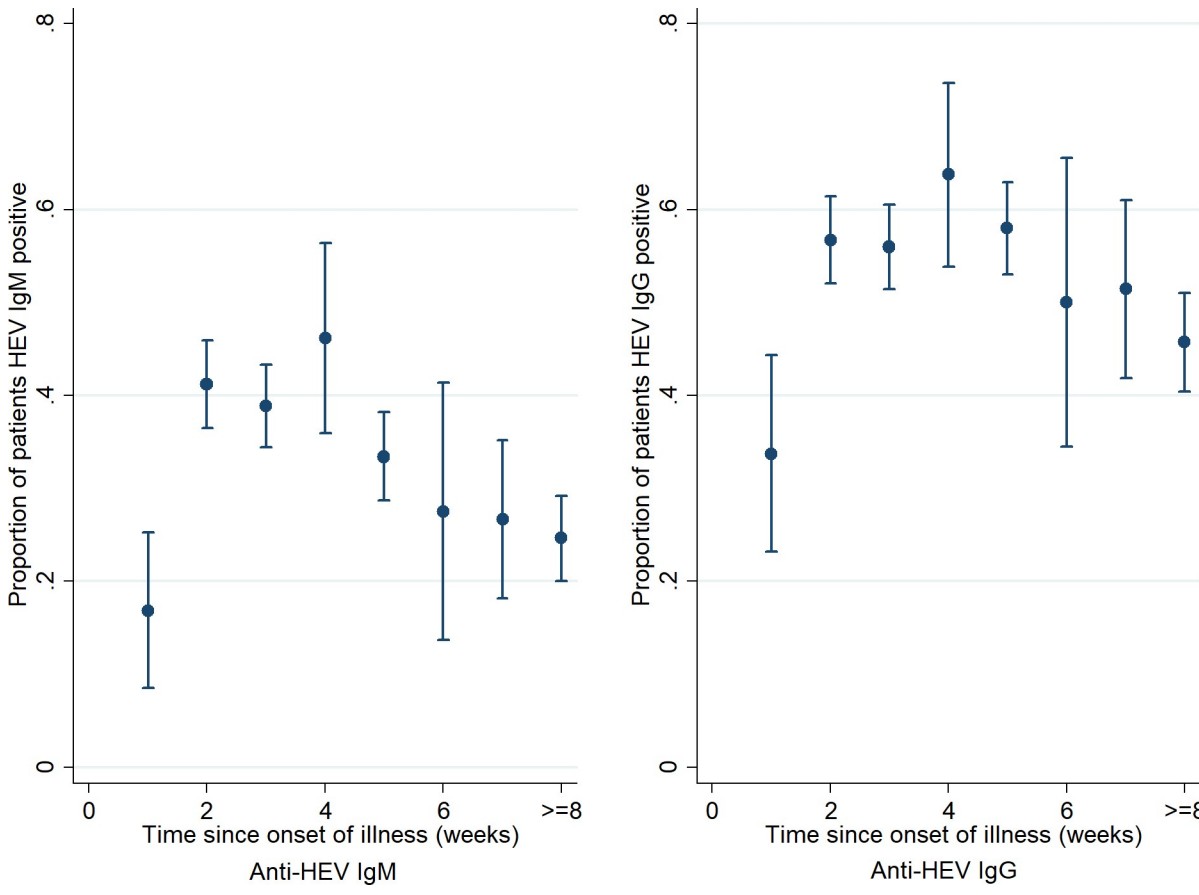

**Fig 2. Proportion of patients positive for anti-HEV IgM and anti-HEV IgG by duration of illness at the time of admission to hospital in six tertiary hospitals in Bangladesh, December 2014- September 2017.**

Patients admitted to hospital with acute HEV infection were detected every month but no specific seasonal pattern was observed during the surveillance period (Fig 3). However, the number of hepatitis E cases generally increased over time.

Most of the signs and symptoms were similar for anti-HEV IgM positive and negative patients (Table 2). However, oedema, distended abdomen, melaena and unconsciousness were observed more often among anti-HEV IgM negative patients than anti-HEV IgM positive patients. The median serum glutamic-pyruvic transaminase level was significantly higher among anti-HEV IgM positive patients than the anti-HEV IgM negative patients [520 (range: 221–1100) IU/L vs. 140 (range: 60–394) IU/L; p <0.001]. We included a list of clinical signs and symptoms of acute jaundice patients that were collected at the time of hospital admission as an appendix (S1 Table).

Of the 115 anti-HEV IgM positive samples that were sent to CDC, 73 (64%) were positive for HEV RNA; none of the anti-HEV IgM negative samples (n = 403) were positive for HEV RNA. In CDC laboratory, all of the samples that tested anti-HEV IgM negative in Bangladesh except one were negative (402 of 403) and all of the anti-HEV IgM samples that tested positive in Bangladesh except two were positive (113 of 115).

## Patient follow-up

Of the 1,925 patients with acute jaundice, we followed up 1,765 (92%), and of them 302 (17%) died. Among those who died, 65 (22%) died during hospitalization and the remaining died

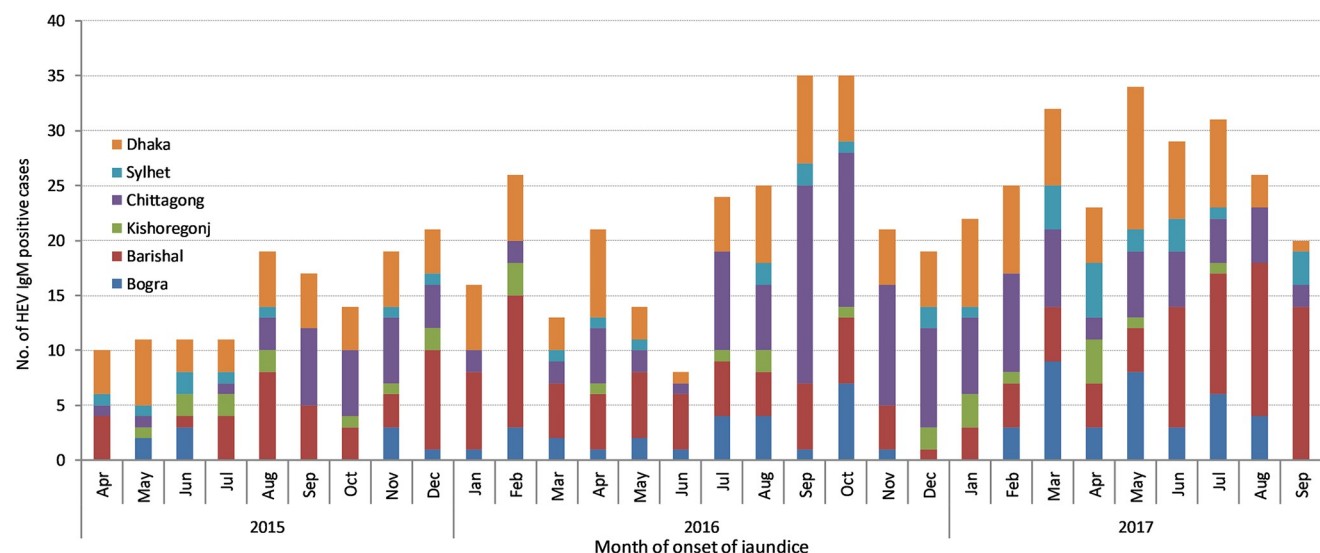

**Fig 3. Number of anti-HEV IgM positive cases by the month of onset of jaundice among patients with acute jaundice admitted in six tertiary hospitals in Bangladesh, April 2015–September 2017.** Note: Since surveillance was established in all hospitals by March, 2015, the seasonality curve covered the period from April 2015 to September 2017.

**Table 2. Signs and symptoms during illness among anti-HEV IgM positive and negative patients admitted in six tertiary hospitals in Bangladesh, December 2014–September 2017.**

| Signs and symptoms during illness | Anti-HEV IgM Positive (N = 661) n (%) | Anti-HEV IgM Negative (N = 1,264) n (%) | P-value[a] |
|---|---|---|---|
| **Examined by physicians** | | | |
| Yellow skin | 660 (100) | 1,258 (100) | 0.263 |
| Yellow eyes | 660 (100) | 1,262 (100) | 0.971 |
| Oedema | 30 (5) | 164 (13) | <0.001 |
| Dehydration | 134 (20) | 237 (19) | 0.425 |
| Distended abdomen | 69 (10) | 228 (18) | <0.001 |
| **Reported by patients/caregivers** | | | |
| Fever | 596 (90) | 1,121 (89) | 0.321 |
| Nausea/vomiting | 602 (91) | 1,079 (85) | <0.001 |
| Anorexia | 536 (81) | 1,020 (81) | 0.862 |
| Abdominal pain | 396 (60) | 779 (62) | 0.450 |
| Melaena/ Clay-colored stools | 107 (16) | 260 (21) | 0.020 |
| Unconsciousness | 37 (6) | 115 (9) | 0.007 |
| **Liver function test result** | | | |
| Serum total bilirubin levels, mg/dl, median(IQR)[b] | 9 (9–13) | 8 (4–15) | 0.063 |
| Patient with above the nomal bilirubin level (Reference level: 0.2–1.2 mg/dl) | 654 (100) | 1,193 (95) | <0.001 |
| Serum Glutamic-Pyruvic Transaminase (SGPT), IU/L, median(IQR) [†] | 520 (221–1100) | 140 (60–394) | <0.001 |
| Patient with above the normal SGPT level (Reference level: 20–60 IU/L) | 613 (94) | 946 (76) | <0.001 |

[a] Chi-squared test comparing anti-HEV IgM result and patients' symptoms during illness

[b] Wilcoxon rank sum test

after leaving hospital but within three months of hospital discharge. Patients with acute HEV infection were significantly less likely to die than the patients who were HEV negative [4.8% (95% CI: 3.2%-6.8%) vs. 23.3% (95% CI: 20.9%-25.8%); *p<0.001*, Table 3). Among the acute hepatitis E patients, case fatality was higher among females than males (9% vs. 3%; *p<0.001*), particularly among pregnant women (12%). HEV infected patients who died were more likely to be aged ≥60 years, have a higher level of serum bilirubin (≥15 mg/dl), have co-infection with HAV or HBV than the HEV infected patients who did not die (Table 3). Fourteen of the 28 deaths in HEV infected patients (50%) and 67 of 274 deaths in HEV uninfected patients (24%) occurred within the first week of hospitalisation.

Among the 192 enrolled pregnant women, we followed up 173 (90%) until the outcome of their pregnancy. Among the 66 mothers who were anti-HEV IgM positive, 53 (80%) had a live birth and among the 107 mothers who were anti-HEV IgM negative, 72 (67%) had a live birth (Table 4). Nineteen percent (10/53) of the live-born babies born to mothers with acute HEV infection died compared to 7% (5/72) of babies born to women without HEV infection (p = 0.038). The median age at death of the babies who were born to HEV infected mothers was two days (IQR: 1–3 days; all within one week) and was five days (IQR: 2–8 days) for those born to HEV uninfected mothers (p = 0.207). Of the 62 HEV infected mothers who were alive until the delivery, 9 (15%) had a miscarriage/abortion or stillbirth.

## Discussion

This study confirms that HEV is a major cause of acute jaundice, is often serious enough to require hospitalization in all regions in Bangladesh, occurs throughout the year, and is associated with considerable mortality, especially among pregnant women and those co-infected with other hepatitis viruses (HAV or HBV). Fifteen percent of HEV infected pregnant women had a miscarriage or stillbirth, and of the children who were born alive, 19% died, all within one week of birth. Three-quarters of the acute hepatitis E cases in hospitals were male, and HEV infection was higher among patients residing in urban areas than patients in rural areas (41% vs 32%).

Previous estimates of hepatitis E prevalence among hospitalized acute jaundice patients in other endemic countries ranges from 10–70% [35,36]. One study in Bangladesh tested 22 fulminant hepatitis patients admitted in a tertiary hospital and detected acute HEV infection among 64% of the patients [37]. Another study in Bangladesh tested 69 retrospectively collected samples from admitted acute-on-chronic liver failure patients in the hepatology unit of a tertiary hospital and identified acute HEV infection among 22% of the samples [34]. The large variation of hepatitis E prevalence might be due to the small number of patients recruited, geographic location of hospitals, and the types of patients recruited. One study that recruited a large number of acute hepatitis patients (685 patients during four years) from the liver clinic of a tertiary hospital in India reported the prevalence of hepatitis E as 39% [43], which is similar to the estimate in our study.

About three-quarters of laboratory-confirmed HEV infected patients were males. One explanation for this may be that males are more involved in outdoor activities than females in low- and middle-income countries [44], putting them at higher risk for exposure to contaminated drinking water. Other reasons for this observation could be due to gender differences in health-seeking behaviour and access to health services. A recent study noted that healthcare expenditure on females is significantly lower than on males in low-income settings [45].

Hepatitis E patients were more likely to be more educated, reside in urban areas and from a high-income group than HEV negative acute jaundice patients. Higher HEV infection rates among educated and high-income group patients might be associated with higher exposure to

**Table 3. Survival status of patients with acute jaundice admitted in six tertiary hospitals in Bangladesh, December 2014-September 2017 (patients followed-up 3 months post hospital discharge).**

| Characteristics | Anti-HEV IgM (+) patients | | | | Anti-HEV IgM (-) patients | | | | All patients | |
|---|---|---|---|---|---|---|---|---|---|---|
| | Patients followed-up | Died | | P-value[a] | Patients followed-up | Died | | P-value[a] | Patients followed-up | Died |
| | | n | Percent (95% CI) | | | n | Percent (95% CI) | | | n (%) |
| *Number of patients* | *589* | *28* | *5 (3–7)* | | *1176* | *274* | *23 (21–26)* | | *1765* | *302 (17)* |
| Sex | | | | 0.001 | | | | 0.07 | | |
| Male | 430 | 13 | 3 (2–5) | | 762 | 165 | 22 (19–25) | | 1192 | 178 (15) |
| Female | 159 | 15 | 9 (5–15) | | 414 | 109 | 26 (22–31) | | 573 | 124 (22) |
| Pregnancy status of women | | | | 0.574 | | | | 0.017 | | |
| Non-Pregnant | 91 | 7 | 8 (3–15) | | 300 | 90 | 30 (25–36) | | 391 | 97 (25) |
| 1st/2nd trimester | 40 | 4 | 10 (3–24) | | 32 | 7 | 22 (9–40) | | 72 | 11 (15) |
| 3rd trimester | 28 | 4 | 14 (4–33) | | 82 | 12 | 15 (8–24) | | 110 | 16 (15) |
| Age-group (years) | | | | <0.001 | | | | <0.001 | | |
| 14–19 | 114 | 4 | 4 (1–9) | | 201 | 17 | 9 (5–13) | | 315 | 21 (7) |
| 20–29 | 263 | 7 | 3 (1–5) | | 313 | 25 | 8 (5–12) | | 576 | 32 (6) |
| 30–39 | 122 | 7 | 6 (2–11) | | 189 | 33 | 18 (12–24) | | 311 | 40 (13) |
| 40–49 | 55 | 5 | 9 (3–20) | | 159 | 55 | 35 (27–43) | | 214 | 60 (28) |
| 50–59 | 27 | 2 | 7 (1–24) | | 143 | 54 | 38 (30–46) | | 170 | 56 (33) |
| 60+ | 8 | 3 | 38 (9–76) | | 171 | 90 | 53 (45–60) | | 179 | 93 (52) |
| Serum total bilirubin levels | | | | 0.001 | | | | <0.001 | | |
| < 5.0 mg/dl | 106 | 4 | 4 (1–9) | | 362 | 52 | 14 (11–18) | | 468 | 56 (12) |
| 5.0–9.9 mg/dl | 233 | 6 | 3 (1–6) | | 320 | 64 | 20 (16–25) | | 553 | 70 (13) |
| 10.0–14.9 mg/dl | 134 | 4 | 3 (1–7) | | 211 | 64 | 30 (24–37) | | 345 | 68 (20) |
| > = 15.0 mg/dl | 116 | 14 | 12 (7–19) | | 283 | 94 | 33 (28–39) | | 399 | 108 (27) |
| Serum glutamic-pyruvic transaminase | | | | 0.082 | | | | 0.001 | | |
| < 200 IU/L | 140 | 12 | 9 (5–15) | | 696 | 181 | 26 (23–29) | | 836 | 193 (23) |
| 200–499 IU/L | 150 | 7 | 5 (2–9) | | 248 | 62 | 25 (20–31) | | 398 | 69 (17) |
| 500–999 IU/L | 133 | 3 | 2 (0–6) | | 126 | 21 | 17 (11–24) | | 259 | 24 (9) |
| > = 1000 IU/L | 162 | 6 | 4 (1–8) | | 104 | 10 | 10 (5–17) | | 266 | 16 (6) |
| Anti-HAV IgM (+) | 8 | 2 | 25 (3–65) | 0.007 | 132 | 2 | 2 (0–5) | <0.001 | 140 | 4 (3) |
| Anti-HBc IgM (+) | 12 | 3 | 25 (5–57) | 0.001 | 258 | 43 | 17 (12–22) | 0.004 | 270 | 46 (17) |
| HBsAg positive (+) | 134 | 12[b] | 9 (5–15) | 0.009 | 487 | 98 | 20 (17–24) | 0.03 | 621 | 110 (18) |
| Duration of illness at the time of admission to hospital | | | | 0.191 | | | | <0.001 | | |
| < 2 weeks | 250 | 8 | 3 (1–6) | | 443 | 65 | 15 (12–18) | | 693 | 73 (11) |
| 2–4 weeks | 203 | 9 | 4 (2–8) | | 344 | 70 | 20 (16–25) | | 547 | 79 (14) |
| 5–6 weeks | 89 | 7 | 8 (3–16) | | 202 | 63 | 31 (25–38) | | 291 | 70 (24) |
| > 6 weeks | 47 | 4 | 9 (2–20) | | 187 | 76 | 41 (34–48) | | 234 | 80 (34) |

[a] Chi-Squared test comparing proportion of deaths by patient characteristics

[b] Three of the HBsAg positive cases were also positive for Anti-HBc IgM

**Table 4. Reported complications during pregnancy, pregnancy outcomes and survival status of newborn babies by anti-HEV IgM test status of mother in six tertiary hospitals in Bangladesh, December 2014–September 2017.**

| Characteristics | Among anti-HEV IgM (+) cases | | Among anti-HEV IgM (-) cases | | P-value[a] |
|---|---|---|---|---|---|
| | n | Percent (95% CI) | n | Percent (95% CI) | |
| **Complications during pregnancy/delivery** | N = 66 | | N = 107 | | |
| Excessive vaginal bleeding | 18 | 27 (18–39) | 48 | 45 (36–54) | 0.021 |
| Convulsions | 7 | 11 (5–20) | 18 | 17 (11–25) | 0.259 |
| Unconscious | 10 | 15 (8–26) | 26 | 24 (17–33) | 0.15 |
| **Pregnancy outcomes** | N = 66 | | N = 107 | | |
| Live birth | 53 | 80 (59–88) | 72 | 67 (58–75) | 0.063 |
| Still birth | 6 | 9 (4–18) | 21 | 20 (13–28) | 0.064 |
| Miscarriage (Spontaneous Abortion) | 2 | 3 (1–10) | 3 | 3 (1–8) | 0.931 |
| Induced abortion | 1 | 2 (0–8) | 5 | 5 (2–10) | 0.270 |
| Patient died before pregnancy outcome | 4 | 6 (2–15) | 6 | 6 (3–12) | 0.901 |
| **Follow-up of live-births** | N = 53 | | N = 72 | | |
| Neonatal deaths | 10 | 19 (11–31) | 5 | 7 (3–15) | 0.038 |

Note: Out of 192 pregnant women, 173 women could be followed-up for pregnancy outcome

[a] Chi-squared test comparing anti-HEV IgM result and different characteristics

contaminated drinking water due to their greater involvement in jobs outside the home as reported in previous studies [12,46]. An alternative explanation might be that children in low socio-economic groups are exposed to pathogens frequently at a young age and so develop robust immunity that protects them from clinically severe illness later in life [47]. This pattern of higher risk of illness among wealthier adolescents and young adults has been observed in some other fecal-oral transmitted diseases, most notably hepatitis A and typhoid fever [48–50]. Higher HEV infection among patients who resided in urban areas might be related to higher possibility of fecal contamination of drinking water sources in urban settings. In large cities in Bangladesh, municipal water pipes are commonly exposed to sewerage lines which may lead to fecal contamination of drinking water [51].

Although males and non-pregnant females who were HEV IgM negative were more likely to die than those with HEV infection, the case fatality in pregnancy was similar between both groups. Furthermore, the occurrence of stillbirths and miscarriages was also common in both HEV and non-HEV acute jaundice patients. However, the neonatal mortality among children born to mothers with HEV infection was significantly higher than the children born to mothers without HEV infection. The higher proportion of neonatal deaths among mother with HEV infection might be explained by the well-described risk of maternal to child transmission of HEV, or the frequency of pre-term delivery [23,25,52]. However, further research is required to understand the contribution of the vertical transmission of HEV to fetal and neonatal mortality, and the mechanism of maternal response to HEV infection.

Although most reported hepatitis E epidemics have been related to fecally contaminated drinking water in the rainy season and hot summer months [13], we did not observe any strong seasonal trend. However, an increase in the number of HEV IgM positive cases was reported from the Chittagong Medical College Hospital during Sep-Oct, 2016 and from the Barishal Medical College Hospital during Jun-Sep, 2017. The unusually high number of cases suggests that there might have been an outbreak in those months which was undetected in real-time, perhaps due to delays in laboratory testing. There were no reported hepatitis E outbreaks in Bangladesh during the surveillance period. A previous study conducted in a private

laboratory in Dhaka identified small outbreaks of HEV infections throughout the year [9]. The patients in our surveillance hospitals might be sporadic cases of hepatitis E or cases from small outbreaks that occurred throughout the year, as was observed in that study [9].

There are a number of limitations of our study. We observed a declining trend in HEV infection positivity by the duration of onset of illness at the time of hospitalization (Fig 2). Anti-HEV IgM antibody titres might have waned for patients who were admitted to hospital in the later stage of their illness [53]; this may lead to an underestimate the rate of HEV infection among acute jaundice patients. In our study, we used Wantai ELISA which is largely accepted as the most sensitive among the available commercial assays [54]. However, the study conducted by Huang et al. observed that about 3% of acute hepatitis cases (defined as acute liver damage evidenced by a 2.5 fold upper limit of the normal level of alanine aminotransferase) were negative for anti-HEV IgM (using the Wantai kit) but had a 4-fold rise in anti-HEV IgG in their convalescent sera [53], which indicates an acute HEV infection [55–57]. Therefore, our results likely underestimate the true prevalence of acute hepatitis E due to the performance of the test kit. HEV infections are generally self-limiting and do not require hospitalization [2] and the patients admitted in tertiary hospitals are severely ill; therefore, the findings in our study represent only the prevalence of hepatitis E among severely ill patients. Finally, in our study, the etiological agent of acute jaundice was unknown for about one-third of the patients, and the samples were not tested for hepatitis C virus.

This study provides a reliable estimate of the prevalence of HEV infection among hospitalized patients with acute jaundice in a hepatitis E endemic country by enrolling patients admitted from a wide geographic area through systematic surveillance over a two and a half year period. It confirms that hepatitis E is the leading cause of acute jaundice in Bangladesh. We identified a high prevalence of hepatitis E among patients with acute jaundice in all of the hospitals included in the study and throughout the multiple ecological zones in Bangladesh. Hepatitis E is therefore a considerable public health problem in Bangladesh. Considering the high burden of HEV among hospitalized patients with acute jaundice, Bangladesh could consider to take control measures to reduce this risk including improvements in water quality, sanitation and hygiene practices and the introduction of hepatitis E vaccine in high-risk areas. An improved understanding of the burden of hepatitis E is required in other endemic countries to plan for effective global actions to prevent HEV infections including the introduction of hepatitis E vaccine.

## Disclaimer

The findings and conclusions in this report are those of the authors and do not necessarily represent the official position of the US Centers for Disease Control and Prevention.

## Supporting information

**S1 Checklist. STROBE checklist.**
(DOCX)

**S1 Table. List of clinical signs and symptoms of acute jaundice patients collected at the time of admission in the surveillance hospitals.**
(DOCX)

**S2 Table. Post hospital discharge follow-up schedule of enrolled patients in the surveillance hospitals.** Enrolled patients and the newborns were followed up post hospital discharge to ascertain their vital status.
(DOCX)

**S3 Table. Hepatitis E serological test results by educational status and residence (rural-urban) of patients with acute jaundice in six tertiary hospitals in Bangladesh, December 2014–September 2017.**
(DOCX)

**S1 Fig. Flow chart of patient enrolment, laboratory testing and follow-up in the surveillance hospitals.** Number of patients diagnosed with acute jaundice, provided a blood specimen and followed up post hospital discharge in the six tertiary hospitals in Bangladesh.
(TIF)

## Acknowledgments

We are grateful to the collaborating hospitals for their interest, enthusiasm and efforts for this study. We gratefully acknowledge the contribution of the study participants and the surveillance medical officers. We are thankful to the field assistants of icddr,b and laboratory staff of IEDCR for their hard work in this study.

## Author Contributions

**Conceptualization:** Repon C. Paul, Stephen P. Luby, Emily S. Gurley.

**Data curation:** Repon C. Paul, Arifa Nazneen, Kajal C. Banik, Shariful Amin Sumon.

**Formal analysis:** Repon C. Paul, Heather F. Gidding, Andrew Hayen.

**Funding acquisition:** Emily S. Gurley.

**Investigation:** Repon C. Paul, Arifa Nazneen, Arifa Akram, Tahir Iqbal, Alexandra Tejada-Strop.

**Methodology:** Repon C. Paul, Saleem Kamili, Stephen P. Luby, Emily S. Gurley.

**Project administration:** Repon C. Paul, Arifa Nazneen, Kajal C. Banik, Kishor K. Paul, M. Salim Uzzaman.

**Supervision:** Repon C. Paul, Heather F. Gidding, Andrew Hayen, Emily S. Gurley.

**Validation:** Repon C. Paul.

**Writing – original draft:** Repon C. Paul.

**Writing – review & editing:** Repon C. Paul, Arifa Nazneen, Kajal C. Banik, Shariful Amin Sumon, Kishor K. Paul, Arifa Akram, M. Salim Uzzaman, Tahir Iqbal, Alexandra Tejada-Strop, Saleem Kamili, Stephen P. Luby, Heather F. Gidding, Andrew Hayen, Emily S. Gurley.

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
