## [Decision Letter · Decision Letter 0]

20 Sep 2019

Dear Mr Paul:

Thank you very much for submitting your manuscript "Hepatitis E as a cause of adult hospitalization in Bangladesh: Results from an acute jaundice surveillance study in six tertiary hospitals, 2014-2017" (PNTD-D-19-01051) for review by PLOS Neglected Tropical Diseases. Your manuscript was fully evaluated at the editorial level and by independent peer reviewers. The reviewers appreciated the attention to an important topic but identified some aspects of the manuscript that should be improved.

We therefore ask you to modify the manuscript according to the review recommendations before we can consider your manuscript for acceptance. Your revisions should address the specific points made by each reviewer.

(1) A letter containing a detailed list of your responses to the review comments and a description of the changes you have made in the manuscript.

(2) Two versions of the manuscript: one with either highlights or tracked changes denoting where the text has been changed (uploaded as a "Revised Article with Changes Highlighted" file ); the other a clean version (uploaded as the article file).

(3) If available, a striking still image (a new image if one is available or an existing one from within your manuscript). If your manuscript is accepted for publication, this image may be featured on our website. Images should ideally be high resolution, eye-catching, single panel images; where one is available, please use 'add file' at the time of resubmission and select 'striking image' as the file type. 

Please provide a short caption, including credits, uploaded as a separate "Other" file. If your image is from someone other than yourself, please ensure that the artist has read and agreed to the terms and conditions of the Creative Commons Attribution License at http://journals.plos.org/plosntds/s/content-license (NOTE: we cannot publish copyrighted images). 

(4) Appropriate Figure Files 

Please remove all name and figure # text from your figure files upon submitting your revision. Please also take this time to check that your figures are of high resolution, which will improve both the editorial review process and help expedite your manuscript's publication should it be accepted. Please note that figures must have been originally created at 300dpi or higher. Do not manually increase the resolution of your files. For instructions on how to properly obtain high quality images, please review our Figure Guidelines, with examples at: http://journals.plos.org/plosntds/s/figures

While revising your submission, please upload your figure files to the Preflight Analysis and Conversion Engine (PACE) digital diagnostic tool, https://pacev2.apexcovantage.com/ PACE helps ensure that figures meet PLOS requirements. To use PACE, you must first register as a user. Then, login and navigate to the UPLOAD tab, where you will find detailed instructions on how to use the tool. If you encounter any issues or have any questions when using PACE, please email us at figures@plos.org.

We hope to receive your revised manuscript by Nov 19 2019 11:59PM. If you anticipate any delay in its return, we ask that you let us know the expected resubmission date by replying to this email.

To submit your revised files, please log in to https://www.editorialmanager.com/pntd/

Sincerely,

Alan L Rothman, MD

Associate Editor

Samuel Scarpino

Deputy Editor

Reviewer's Responses to Questions

**Key Review Criteria Required for Acceptance?**

**Methods**

-Are the objectives of the study clearly articulated with a clear testable hypothesis stated?

-Is the study design appropriate to address the stated objectives?

-Is the population clearly described and appropriate for the hypothesis being tested?

-Is the sample size sufficient to ensure adequate power to address the hypothesis being tested?

-Were correct statistical analysis used to support conclusions?

-Are there concerns about ethical or regulatory requirements being met?

Reviewer #1: The objectives were clear and the method appropriate. There was a robust sample size well distributed throughout the country to represent a national burden for severe hospitalized jaundice. As acknowledged by the authors, the design did not allow for estimates of milder disease, or those with severe disease but seeking traditional medicine.

Reviewer #2: The objective of this study was to estimate the prevalence of HEV infection among jaundice patients aged >=14 ansadmitted in tertiary hospitals in Bangladesh, estimate the case-fatality rate and estimate factors associate with mortality, as well as describe seasonal and geographical trends

The descrition of objective is very complex. Which is the main objective of the study? Please rewrite this section and specify the main objective.

The study is well designed, but please insert the list of items included in the case report (line 138) and a schema of the method of enrollement of patients

The sample size was not reported in the method section, please add 

The discussion and conclusion were based on univariate analysis. Please add multivariate analysis to validate results or univariate analysis

Please add number of ethic autorisation

**Results**

-Does the analysis presented match the analysis plan?

-Are the results clearly and completely presented?

-Are the figures (Tables, Images) of sufficient quality for clarity?

Reviewer #1: The results are clearly represented. It seems that all causes of jaundice have a negative impact on pregnancies. Although there were more neonatal deaths with HEV + pregnant women, maternal mortality and miscarriage were equally bad among HEV+ and HEV- pregnant women. This might have been brought up in the discussion more fully. Hepatitis is bad for maternal and fetal outcomes regardless the type.

Reviewer #2: As mentioned in method section authors have to add a multivariate analysis in order to validate results of the univariate analysis

Please add median age and Min and Max in Table 1

In Figure 2 authors declared that the proportions of patients anti-HEV IgM declined significatly with time after two months but IC95% from the week 2 to four overlap. Please modify this sentence.

Figure 3: How do you explain the increase of the number of IgM positive cases during the 2016 summer? 

Line 304-305 please place after line 301

Table 4: pvalue of live birth/still birth are missed?

**Conclusions**

-Are the conclusions supported by the data presented?

-Are the limitations of analysis clearly described?

-Do the authors discuss how these data can be helpful to advance our understanding of the topic under study?

-Is public health relevance addressed?

Reviewer #1: The conclusions are supported by the evidence and the limitations nicely acknowledged.

Reviewer #2: The authors need to improve statistical analysis

**Editorial and Data Presentation Modifications?**

Reviewer #1: none

Reviewer #2: (No Response)

**Summary and General Comments**

Reviewer #1: This is an important paper which well describes the burden of HEV among severe jaundice cases seeking hospital care in Bangladesh. While the lack of quality burden of disease data is an important reason why there is not clear and strong guidance regarding use of the HEV vaccine, it is not the only limitation. There is insufficient experience with the vaccine in high risk populations , like pregnant women or those in humanitarian emergency settings or other areas (like Bangladesh) where genotype 1 and fecal-oral transmission is the common route. 

It is unfortunate that those under 14 yo and hospitalized with jaundice could not be included, or that families were not investigated to find additional mild or asymptomatic cases. But I understand the limitations of funding. Hopefully these important results will lead to additional studies to extend the understanding of HEV transmission into the household and to younger ages. If vaccines are to be used most effectively, the entire transmission cycle should be known.

Reviewer #2: (No Response)

PLOS authors have the option to publish the peer review history of their article (what does this mean?). If published, this will include your full peer review and any attached files.

Reviewer #1: No

Reviewer #2: No

---

## [Editor Report · Decision Letter 1]

27 Dec 2019

Dear Mr Paul,

We are pleased to inform you that your manuscript, "Hepatitis E as a cause of adult hospitalization in Bangladesh: Results from an acute jaundice surveillance study in six tertiary hospitals, 2014-2017", has been editorially accepted for publication at PLOS Neglected Tropical Diseases.

Before your manuscript can be formally accepted and sent to production you will need to complete our formatting changes, which you will receive in a follow up email. Please note: your manuscript will not be scheduled for publication until you have made the required changes.

IMPORTANT NOTES

* Copyediting and Author Proofs: To ensure prompt publication, your manuscript will NOT be subject to detailed copyediting and you will NOT receive a typeset proof for review. The corresponding author will have one final opportunity to correct any errors when sent the requests mentioned above. Please review this version of your manuscript for any errors.

* If you or your institution will be preparing press materials for this manuscript, please inform our press team in advance at plosntds@plos.org. If you need to know your paper's publication date for media purposes, you must coordinate with our press team, and your manuscript will remain under a strict press embargo until the publication date and time. PLOS NTDs may choose to issue a press release for your article. If there is anything that the journal should know, please get in touch.

*Now that your manuscript has been provisionally accepted, please log into EM and update your profile. Go to http://www.editorialmanager.com/pntd, log in, and click on the "Update My Information" link at the top of the page. Please update your user information to ensure an efficient production and billing process.

*Note to LaTeX users only - Our staff will ask you to upload a TEX file in addition to the PDF before the paper can be sent to typesetting, so please carefully review our Latex Guidelines [http://www.plosntds.org/static/latexGuidelines.action] in the meantime.

Best regards,

Alan L Rothman, MD

Associate Editor

Samuel Scarpino

Deputy Editor

---

## [Editor Report · Acceptance letter]

13 Jan 2020

Dear Mr Paul,

We are delighted to inform you that your manuscript, "Hepatitis E as a cause of adult hospitalization in Bangladesh: Results from an acute jaundice surveillance study in six tertiary hospitals, 2014-2017," has been formally accepted for publication in PLOS Neglected Tropical Diseases.

Best regards,

Serap Aksoy

Editor-in-Chief

Shaden Kamhawi

Editor-in-Chief
